# Automated Multi-level Preference for MLLMs

**Mengxi Zhang**[1,2], **Wenhao Wu**[3], **Yu Lu**[4], **Yuxin Song**[1], **Kang Rong**[1], **Huanjin Yao**[1,5]
**Jianbo Zhao**[6], **Fanglong Liu**[1], **Haocheng Feng**[1], **Jingdong Wang**[1], **Yifan Sun**[1*]
[1]Baidu Inc.    [2]Tianjin University    [3]The University of Sydney
[4]University of Technology Sydney    [5]Tsinghua University    [6]Chinese Academy of Science

## Abstract

Current multimodal Large Language Models (MLLMs) suffer from "hallucination", occasionally generating responses that are not grounded in the input images. To tackle this challenge, one promising path is to utilize reinforcement learning from human feedback (RLHF), which steers MLLMs towards learning superior responses while avoiding inferior ones. We rethink the common practice of using binary preferences (*i.e.*, superior, inferior), and find that adopting multi-level preferences (*e.g.*, superior, medium, inferior) is better for two benefits: 1) It narrows the gap between adjacent levels, thereby encouraging MLLMs to discern subtle differences. 2) It further integrates cross-level comparisons (beyond adjacent-level comparisons), thus providing a broader range of comparisons with hallucination examples. To verify our viewpoint, we present the Automated Multi-level Preference (**AMP**) framework for MLLMs. To facilitate this framework, we first develop an automated dataset generation pipeline that provides high-quality multi-level preference datasets without any human annotators. Furthermore, we design the Multi-level Direct Preference Optimization (MDPO) algorithm to robustly conduct complex multi-level preference learning. Additionally, we propose a new hallucination benchmark, MRHal-Bench. Extensive experiments across public hallucination and general benchmarks, as well as our MRHal-Bench, demonstrate the effectiveness of our proposed method. Code is available at `https://github.com/takomc/amp`.

## 1  Introduction

Multimodal Large Language Models (MLLMs) [1, 2, 3, 4, 5, 6] have achieved remarkable advancement in vision-language understanding tasks, *e.g.*, vision question answering [7], image captioning [8], and human-machine conversation. Despite MLLMs achieving significant breakthroughs, they still suffer from hallucinations [9, 10], referring to responses that are not accurately anchored to the context provided by images. This problem shrinks the performance of MLLMs and draws considerable research attention. To mitigate the hallucinations, some existing methods [11, 12, 13, 14] adopt Reinforcement Learning from Human Feedback (RLHF) methods, which collect human/AI preferences and integrate them into the MLLMs optimization process via reinforcement learning.

Existing RLHF methods have demonstrated that comparing superior and inferior responses within a binary-level preference framework can improve the performance of optimized MLLMs. However, *Is a single comparison between superior and inferior responses sufficient for preference learning in MLLMs?* Upon consideration, we find that a multi-level preference framework offers greater benefits for preference learning, primarily due to two main intuitive advantages. Firstly, reducing the gap between adjacent levels helps mitigate the challenge of distinguishing micro hallucinations in responses. As depicted in Fig. 1, in the baseline method (*i.e.*, binary preference), significant differences exist between the superior response A and the inferior response C. By introducing an additional medium response and shifting the focus to multiple comparisons between adjacent levels

---

*Corresponding author

38th Conference on Neural Information Processing Systems (NeurIPS 2024).

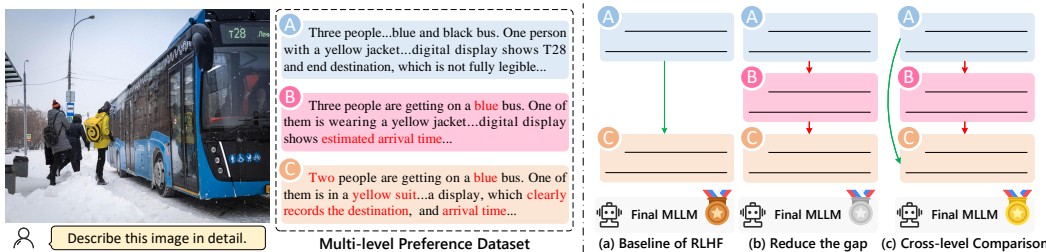

Figure 1: **Left:** Depicted are the input image, text prompt, and corresponding multi-level preference dataset. Contents highlighted in red signify hallucinations. Responses range from A to C, representing varying degrees of quality from superior to inferior. **Right:** Illustrating the strategy for leveraging inferior responses. (a) displays the conventional RLHF baseline, which adpots the binary-level preference. (b) To mitigate the gap between adjacent levels, we first split a single comparison into multiple comparisons by inserting extra medium responses. (c) Furthermore, we introduce the cross-level comparison to augment the dataset with more hallucination examples.

("A>B", "B>C"), as highlighted by the red arrows in Fig. 1(b), we mitigate this issue. Interestingly, under certain conditions, the MLLM's performance with "A>C" comparison is even worse than that with "B>C", indicating that reducing the gap between adjacent levels is sometimes more effective than enhancing the quality of superior responses. Secondly, cross-level comparisons can further enhance performance. In the comparisons between adjacent levels, the only comparison utilizing the inferior response A is "A>B", which may lead the model to focus more on suppressing hallucinations in response B. To address this, we introduce the cross-level comparison "A>C" (the green arrow in Fig. 1(c)) to provide more negative examples, thereby helping the model suppress more possible hallucinations. By integrating these strategies, we evolve the conventional binary-level preference learning into a more sophisticated multi-level preference learning framework.

However, exploring multi-level preferences for MLLMs poses significant challenges: 1) Labeling multi-level preference datasets is expensive and laborious. While some methods [11, 12] utilize human annotators to obtain preference labels, this approach is effective for binary datasets but falls short for multi-level preference datasets. Specifically, establishing a $K$-level preference dataset requires human annotators to make $K(K-1)/2$ comparisons. For example, with $K = 5$, this results in 10 comparisons, significantly more than is required for binary datasets. On the other hand, datasets annotated by humans or AI often contain significant noise and bias. To investigate this, we collected preferences from both humans and GPT-4V [15] on a subset of ShareGPT4V [16], using three MLLMs to generate varied responses. Setting $K$ to 3, we compared pairs of responses (A&B, B&C, A&C) through three comparisons. However, we observed a frequent contradictory pattern (A>B, B>C, C>A), with rates of approximately 14% and 11% in human and GPT-4V annotations, respectively, resulting in a low-quality multi-level preference dataset. 2) The optimal multi-level preference learning objective remains unclear. While multi-level preference is more beneficial for optimizing MLLMs, it introduces greater complexity than binary preference. Therefore, it requires an effective algorithm to fully utilize the knowledge embedded within multi-level preference datasets.

To overcome the challenges outlined above, we introduce innovative strategies at both the data and method levels: 1) At the data level, we propose two novel methods for generating initial multi-level preference datasets without human or AI annotators. Furthermore, we implement an auto-check mechanism to further refine these datasets by evaluating the scores and accuracy of the generated responses. 2) At the method level, we introduce the Multi-level Direct Preference Optimization (MDPO) algorithm, a derivative of the traditional Direct Preference Optimization (DPO) algorithm [17]. The MDPO algorithm extends the capabilities of the DPO algorithm to facilitate multi-level preference optimization. Additionally, we incorporate a tailored penalty term into the MDPO learning objective to ensure robust multi-level preference learning. 3) Finally, we introduce a new evaluation benchmark, MRHal-Bench, which is the first designed specifically to evaluate hallucinations in multi-round dialogues. In summary, our contributions are as follows:

- Contrary to prior RLHF studies that focused solely on enhancing the quality of superior responses, our findings indicate that inferior responses can also play a crucial role in reducing hallucinations under the multi-level preference learning framework.

- To support effective multi-level preference learning, we develop two novel methods and an auto-check mechanism, enabling the creation of high-quality multi-level preference datasets without the need for human or AI annotators. Furthermore, we design the Multi-level Direct Preference Optimization (MDPO) algorithm with a specifically crafted learning objective, allowing MLLMs to robustly learn from the multi-level preference dataset.

- Our extensive experiments across various hallucination benchmarks confirm the effectiveness of our framework. Additionally, we have introduced MRHal-Bench, the first benchmark specifically designed to evaluate hallucinations in multi-round dialogues.

## 2 Related Work

### 2.1 Multimodal Large Language Models

Recently, the multimodal learning community has witnessed the great success of MLLMs [1, 2, 3, 4, 5, 6, 18], which employ a cross-modal alignment module to connect the visual encoder [19, 20, 21] and the language model [22, 23]. Typically, MLLMs undergo a standard training strategy involving two stages. First, to bridge the gap between visual and textual representations, the cross-modal alignment module is trained on a large-scale multimodal dataset [1, 24, 25], which endows the LLMs with visual-understanding ability. Then, MLLMs are further fine-tuned on specific visual instruction datasets [2, 16, 18, 26] to facilitate various downstream tasks [7, 8]. Despite the significant advancement, MLLMs still suffer from hallucinations, which decrease their performance on multiple tasks and attract increasing attention from researchers.

### 2.2 Hallucinations in MLLMs

Hallucinations in MLLMs [9, 10] denote inconsistencies between the input image and the generated response. Unlike hallucinations in LLMs [27, 28], those observed in MLLMs are more complicated, which attracts more attention from researchers. Some methods [26, 29] focus on reducing hallucinations by constructing high-quality datasets, while others employ specialized mechanisms such as decoding strategies [30, 31], retrieval augmented generation [32], and chain-of-thought [33] to mitigate hallucinations. However, due to the inherent limitations of cross-entropy loss, these methods may provide insufficient guidance for modality alignment. Recently, reinforcement learning-based methods [11, 12, 13, 14, 34], leveraging techniques like DPO [17] and PPO [35], have emerged as promising direction. Yet, these methods rely on preference datasets annotated by humans or AI, which are costly and susceptible to noise. Besides, they follow the traditional binary-level preference framework, which is insufficient for preference learning of MLLMs. To address these problems, we propose a novel AMP framework, utilizing a human-free multi-level preference dataset and the MDPO algorithm to guide MLLMs.

## 3 Methods

In this section, we delve into the Automated Multi-level Preference (AMP) framework. Initially, we outline two strategies for constructing an initial multi-level preference dataset, aligning with two perspectives of the scaling law [36, 37]. Subsequently, we introduce the auto-check mechanism aimed at refining the initial dataset based on relevant metrics. Lastly, we introduce the Multi-level Direct Preference Optimization (MDPO) algorithm, featuring a novel and robust learning objective.

### 3.1 Human-free Multi-level Preference Dataset Generation

The quality of the preference dataset significantly influences the refined model's performance. Constructing a high-quality initial preference dataset relies on two fundamental principles. Firstly, the ranking between superior and inferior responses should be correct in most cases. Secondly, the language style among different responses is expected to be consistent. Specifically, inconsistent language styles can introduce biases that mislead the MLLM, resulting in reward hacking and performance degradation [12, 36]. Considering these factors, we propose the ***Multi-size Expert Generation (MEG)*** and ***Incremental Generation (IG)*** strategies to build reliable preference datasets from the perspectives of model size and dataset size, respectively.

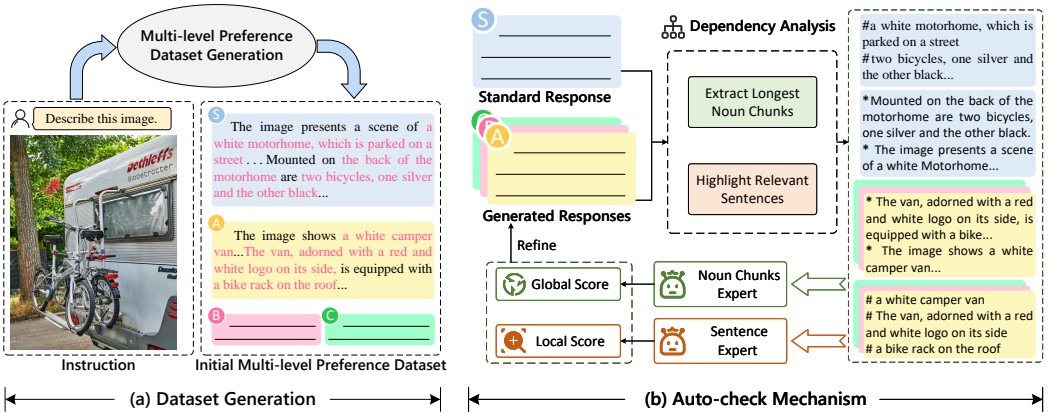

Figure 2: Pipeline for Constructing Human-free Multi-level Preference Dataset. We initiate the process with *Multi-size Expert Generation* and *Incremental Generation* to establish the initial dataset. Then, to enhance the quality of the initial preference dataset, we introduce the *Auto-check Mechanism*, which calculates both global and local metrics based on sentences and noun chunks, respectively.

### 3.1.1 Multi-size Expert Generation

Scaling laws suggest that the performance of the model improves as the model size increases. Thus, a logical strategy is to generate various responses using models of different sizes. For consistency in language style, it's preferable that these models stem from the same family. Specifically, we adpot LLaVA-based models, such as LLaVA-2B [38], LLaVA-7B, LLaVA-13B, and LLaVA-34B [3]. Leveraging the standard response in the instruction tuning dataset [29], we procure up to 5 responses of differing quality.

### 3.1.2 Incremental Generation

In *Multi-size Expert Generation*, our focus lies on employing models of various sizes, while *Incremental Generation* involves training datasets of different sizes. In practice, we partition the entire fine-tuned dataset $\mathcal{F} = \{\mathcal{I}; P; R\}$ into $K - 2$ equal parts for the $K$-rank preference dataset, where $\mathcal{I}$, $P$, and $R$ symbolize the image, text prompt, and standard response, respectively. Then, we use subsets $\mathcal{S}_i = [\mathcal{F}_1, \mathcal{F}_2, ..., \mathcal{F}_i]$ to fine-tune the pre-trained MLLM $\mathcal{M}$, yielding $K - 2$ fine-tuned MLLMs, where $i \in [1, K - 2]$. Hence, the $K - 2$ responses generated by fine-tuned MLLMs, along with the response generated by the pre-trained MLLM and the standard response constitute the $K$-rank preference dataset. The entire process is documented in Algorithm 1.

---

**Algorithm 1** The Pseudocode of Incremental Generation for $K$-rank Preference Dataset.

---

**Input:** Image $\mathcal{I}$, text prompt $P$, and standard response $R$ for fine-tuned dataset $\mathcal{F} = \{\mathcal{I}; P; R\}$, annotated dataset $\mathcal{A} = \{\mathcal{I}_a; P_a; R_a\}$, pre-trained MLLM $\mathcal{M}$.
**Output:** $K$-level preference dataset $\mathcal{D} = \{\mathcal{I}_a; P_a; [R_0, R_1, ..., R_{K-1}]\}$.
1: Split $\mathcal{F}$ into $K - 2$ equal parts $[\mathcal{F}_1, \mathcal{F}_2, ..., \mathcal{F}_{K-2}]$;
2: **for** $(i = 1$ to $K - 2)$ **do**
3:     Train $\mathcal{M}$ with subset $\mathcal{S}_i = [\mathcal{F}_1, \mathcal{F}_2, ..., \mathcal{F}_i] \Rightarrow$ Get fine-tuned MLLM $\mathcal{M}_i$;
4:     $R_i = \mathcal{M}_i(\mathcal{I}_a, P_a)$;     {Generate response $R_i$ via fine-tuned MLLM $\mathcal{M}_i$}
5: **end for**
6: $R_0 = R_a$, $R_{K-1} = \mathcal{M}(\mathcal{I}_a, P_a)$;
7: **return** $\mathcal{D} = \{\mathcal{I}_a; P_a; [R_0, R_1, ..., R_{K-1}]\}$.

---

### 3.1.3 Auto-check Mechanism

In the aforementioned process, we devised two strategies to establish the initial multi-level preference dataset. While the rankings in this dataset are generally accurate, occasional anomalies may lead to incorrect preferences. To enhance the ranking accuracy, we introduce the auto-check mechanism.

First, we identify all nouns in the various responses, including terms like "motorhome", "street", *etc*. Note that certain nouns are deprecated (further details are provided in Appendix A.1). Next, we analyze the dependency relationships within the sentence to extend each noun into the longest possible noun chunks. For example, "a white motorhome, which is parked on a street" would be represented (denoted by pink color in Fig. 2).

After extracting all noun chunks, we send them into the noun chunk expert (*i.e.*, the text encoder of CLIP [19]) to obtain text features $F_S = \{f_{S_1}, f_{S_2}, ..., f_{S_M}\}$ and $F_G = \{f_{G_1}, f_{G_2}, ..., f_{G_N}\}$, where M and N denote the number of noun chunks in standard and generated responses, respectively. We then calculate the similarity score as outlined in Eq. 1:

$$\mathbf{S}[m, n] = \frac{f_{S_m} \cdot f_{G_n}}{\|f_{S_m}\| \times \|f_{G_n}\|}, \qquad \mathbf{s}[m] = max(\mathbf{S}[m, :]), \tag{1}$$

where $\mathbf{S} \in \mathbb{R}^{M \times N}$ is the similarity matrix between standard and generated responses. $\mathbf{s} \in \mathbb{R}^M$ represents the similarity score of generated response $F_G$. We further introduce the accuracy metric:

$$\mathbf{p}[m] = \begin{cases} 1 & \text{if} \quad \mathbf{s}[m] > \tau \\ 0 & \text{otherwise} \end{cases}, \qquad \text{Acc} = \text{Sum}(\mathbf{p})/\text{M}, \tag{2}$$

where $\tau$ is the threshold, set to 0.85. Accuracy (Acc) reflects the completeness of the credible components within the generated response.

While noun chunks represent the consistency at a local level, entire sentences represent global consistency, such as the relationships between multiple objects and the actions of objects, *etc*. To assess global consistency, we retrieve the sentences where each noun chunk is located as the global representation. The relevant metrics of sentences are the same as those of noun chunks.

Finally, we compute the final accuracy and scores by averaging the local and global metrics. Among the generated responses, the one with the highest accuracy is regarded as the best. In cases where multiple responses achieve equal accuracy, the one with the highest scores is considered superior.

### 3.2 Multi-level Direct Preference Optimization (MDPO)

Reinforcement learning algorithms [11, 12, 13, 14, 34] have demonstrated promising results in training MLLMs with human-preference datasets. Encouraged by the success of these pioneers, we delve deeper into the potential of multi-level preferences. In this section, we design the Multi-level Direct Preference Optimization (MDPO) algorithm, furnishing a novel and robust learning objective.

### 3.2.1 Preliminary

Prevalent methods [11, 39, 40] leverage the Proximal Policy Optimization (PPO) algorithm to align with preference data. However, the performance of this approach highly depends on the extra reward model, which is sensitive to noises within the preference dataset. Besides, the last stage of PPO fine-tunes the actor and critic model with the online strategy, resulting in high computational costs and unstable procedures. To mitigate these issues, DPO [17] excludes the reward model by analytically expressing reward functions with optimal policy $\pi_*$ and initial policy $\pi_{\text{ref}}$. Denote $x$ and $y$ as the inputs and outputs of MLLMs, the reward function is converted into:

$$r(x, y) = \beta \log \frac{\pi_*(y|x)}{\pi_{\text{ref}}(y|x)} + \beta \log Z(x), \tag{3}$$

where $Z(\cdot)$ is the partition function, $\beta$ is a constant. Under the Bradley-Terry model, the policy objective becomes:

$$\mathcal{L}_{\text{DPO}}(\pi_\theta; \pi_{\text{ref}}) = -\mathbb{E}_{(x, y_w, y_l) \sim \mathcal{D}} \left[ \log \sigma \left( r(x, y_w) - r(x, y_l) \right) \right],$$
$$= -\mathbb{E}_{(x, y_w, y_l) \sim \mathcal{D}} \left[ \log \sigma \left( \beta \log \frac{\pi_\theta(y_w|x)}{\pi_{\text{ref}}(y_w|x)} - \beta \log \frac{\pi_\theta(y_l|x)}{\pi_{\text{ref}}(y_l|x)} \right) \right], \tag{4}$$

where $\sigma(\cdot)$ represents the Sigmoid function, and $x$, $y_w$, and $y_l$ denote inputs, superior and inferior responses, respectively. In practice, $\pi_{\text{ref}}$ remains frozen during DPO training. Thus, only the policy model $\pi_\theta$ is updated in the training process, ensuring efficiency and cost-effectiveness.

### 3.3 Learning Objective of MDPO Algorithm

To facilitate the multi-level preference dataset, we revise the learning objective for $K$-rank with $K(K-1)/2$ comparisons:

$$
\begin{aligned}
\mathcal{L}_{\mathrm{MDPO}}(\pi_\theta; \pi_{\mathrm{ref}}) &= -\sum_{i=0}^{K-1}\sum_{j=i+1}^{K-1} \mathcal{L}_{\mathrm{DPO}(x,y_i,y_j)\sim\mathcal{D}}, \\
&= -\sum_{i=0}^{K-1}\sum_{j=i+1}^{K-1} \mathbb{E}_{(x,y_i,y_j)\sim\mathcal{D}}\left[\log\sigma\left(\beta\log\frac{\pi_\theta(y_i|x)}{\pi_{\mathrm{ref}}(y_i|x)} - \beta\log\frac{\pi_\theta(y_j|x)}{\pi_{\mathrm{ref}}(y_j|x)}\right)\right],
\end{aligned}
\tag{5}
$$

where the quality of response $y_i$ is superior to $y_j$.

During MDPO training, we observed that despite the loss decreasing normally, the optimized MLLM sometimes generates certain words or phrases repetitively. This occurs because the probability of the policy model producing both superior and inferior responses simultaneously decreases. While the probability of generating inferior responses declines more rapidly, the policy model's capability to generate superior responses also diminishes, leading to an overall deterioration in performance. To mitigate this risk, we introduce an additional penalty term, modifying Eq. 4 as follows:

$$
\mathcal{L}_{\mathrm{DPO-P}}(\pi_\theta; \pi_{\mathrm{ref}}) = -\mathbb{E}_{(x,y_w,y_l)\sim\mathcal{D}}\left[\log\sigma\left(\beta\log\frac{\pi_\theta(y_w\mid x)}{\pi_{\mathrm{ref}}(y_w\mid x)} - \beta\log\frac{\pi_\theta(y_l\mid x)}{\pi_{\mathrm{ref}}(y_l\mid x)}\right) + \log\frac{\pi_\theta(y_w\mid x)}{\pi_{\mathrm{ref}}(y_w\mid x)}\right].
\tag{6}
$$

With this penalty term, the probability of generating superior responses is explicitly improved. To minimize the impact of medium-quality responses, we apply the penalty term exclusively to the best response. Consequently, the learning objective of MDPO is formulated as:

$$
\mathcal{L}_{\mathrm{MDPO}}(\pi_\theta; \pi_{\mathrm{ref}}) = -\left[\sum_{j=1}^{K-1}\mathcal{L}_{\mathrm{DPO-P}(x,y_0,y_j)\sim\mathcal{D}} + \sum_{i=1}^{K-1}\sum_{j=i+1}^{K-1}\mathcal{L}_{\mathrm{DPO}(x,y_i,y_j)\sim\mathcal{D}}\right].
\tag{7}
$$

## 4 Experiments and Analysis

### 4.1 Implementation Details

We adpot LLaVA-v1.5 [3] as our base model for all experiments, which is built upon Vicuna [22, 23] and utilizes ViT-L/14 [19] as the image encoder. Our training dataset contains 1k detailed captions from ShareGPT4V [16], 4k image-text pairs from [34], 4k human-annotated data from [12] and 2k multi-round dialogues annotated by us (the annotated process is detailed in Appendix A.2), forming a total of 11k training instances. For training MDPO, we employ the AdamW [41] optimizer for 4 epochs and apply a peak learning rate of $5 \times 10^{-5}$ with the cosine decay strategy. To enhance learning efficiency, we incorporate LoRA-based [42] fine-tuning, with a low-rank r set to 64 for both attention and feed-forward modules. All experiments are conducted with a batch size of 16 on 8 Nvidia A100 GPUs with 40G VRAM. Further implementation details of the Human-free Multi-level Preference Dataset generation are provided in Appendix A.3.

### 4.2 Evaluation Benchmarks

To verify the effectiveness of our proposed AMP framework, we conduct comprehensive comparisons with various baselines across several benchmarks, including QA-based hallucination benchmark POPE [9], fine-grained hallucination benchmark MMHal-Bench [11], general benchmark LLaVA-Bench [2], and our newly developed multi-round dialogue hallucination benchmark MRHal-Bench. Specifically, POPE assesses the object existence hallucinations by prompting MLLMs to provide

Table 1: Comparison of conventional MLLMs and RLHF-based MLLMs across MMHal-Bench, MRHal-Bench, and LLaVA-Bench. "MEG" represents training data generated via Multi-size Expert Generation, while "IG" indicates training data produced using Incremental Generation.

| Methods | MMHal-Bench | | MRHal-Bench | | LLaVA-Bench | | |
| | Score↑ | Hal.↓ | Score (c/m)↑ | Hal. (c/m)↓ | Conv.↑ | Detail↑ | Comp.↑ |
| --- | --- | --- | --- | --- | --- | --- | --- |
| LLaVA$_{13B}$ [2] | 1.11 | 0.84 | 3.01 / 3.01 | 0.40 / 0.37 | 85.4 | 74.3 | 96.3 |
| InstructBLIP$_{7B}$ [4] | 1.80 | 0.62 | 3.00 / 3.00 | 0.39 / 0.38 | 83.2 | 67.6 | 90.6 |
| LLaVA-v1.5$_{7B}$ [3] | 2.01 | 0.61 | 3.38 / 3.39 | 0.32 / 0.29 | 80.2 | 75.9 | 89.2 |
| DeepSEEK-VL [44] | 2.22 | 0.56 | 3.54 / 3.53 | 0.29 / 0.25 | 74.4 | 76.5 | 78.2 |
| LLaVA-V1.6 [3]$_{7B}$ | 2.30 | 0.59 | 3.80 / 3.78 | 0.27 / 0.26 | 82.3 | 85.3 | 96.9 |
| MiniCPM-V [45] | 2.34 | 0.50 | 3.31 / 3.31 | 0.39 / 0.34 | 80.8 | 75.6 | 89.2 |
| LLaVA-v1.5$_{13B}$ [3] | 2.44 | 0.53 | 3.58 / 3.59 | 0.29 / 0.27 | 81.6 | 75.5 | 95.2 |
| Qwen-VL-Chat [6] | 2.70 | 0.46 | 3.71 / 3.68 | 0.27 / 0.21 | 81.9 | 77.1 | 92.3 |
| LLaVA-V1.6 [3]$_{13B}$ | 3.04 | 0.43 | 3.73 / 3.79 | 0.30 / 0.25 | 89.2 | 90.3 | 98.3 |
| LLaVA-RLHF$_{7B}$ [11] | 2.04 | 0.68 | 3.58 / 3.56 | 0.34 / 0.29 | 85.3 | 74.7 | 105.6 |
| LLaVA-RLHF$_{13B}$ [11] | 2.53 | 0.57 | 3.26 / 3.27 | 0.45 / 0.38 | 93.8 | 74.3 | **111.4** |
| RLHF-V [12] | 2.66 | 0.52 | 2.54 / 2.60 | 0.52 / 0.56 | 93.1 | 75.3 | 91.6 |
| POVID [14] | 2.69 | – | 3.46 / 3.47 | 0.28 / 0.28 | 75.7 | 75.2 | 89.5 |
| SILKIE [34] | 3.02 | – | 3.71 / 3.70 | 0.30 / 0.29 | 86.3 | 76.4 | 95.3 |
| FGAIF [13] | 3.09 | 0.36 | 3.77 / 3.79 | 0.30 / 0.31 | **98.2** | **93.6** | 110.0 |
| AMP-MEG$_{7B}$ | 3.17 | 0.35 | 4.07 / 4.06 | 0.20 / 0.15 | 89.7 | 89.1 | 98.8 |
| AMP-MEG$_{13B}$ | **3.23** | **0.34** | **4.21** / **4.21** | **0.15** / **0.11** | 94.4 | 91.2 | 95.6 |
| AMP-IG$_{7B}$ | 3.12 | 0.41 | 4.02 / 4.04 | 0.22 / 0.13 | 90.2 | 85.9 | 99.8 |
| AMP-IG$_{13B}$ | 3.18 | 0.36 | 3.96 / 4.01 | 0.22 / 0.20 | 91.3 | 86.8 | 99.4 |

Hal.: Hallucination rate, Conv.: Conversation, Detailed: Detailed Description, Comp.: Complex Question, c/m: cumulative/mean.

binary responses ('yes' or 'no'). MMHal-Bench is designed to quantify hallucinations with the assistance of GPT-4 [43]. Different from the simple questions in conventional benchmarks, MMHal-Bench contains more general, open-ended, and fine-grained questions. LLaVA-Bench serves as a general benchmark for systematic comprehension, encompassing three categories: conversation, detailed description, and complex questions. In addition to these benchmarks, we introduce MRHal-Bench to evaluate hallucinations in multi-round dialogues, covering six aspects: attribute, description, existence, counting, reasoning, and spatial relation. For further details, please refer to Appendix A.4.

### 4.3 Comparisons with Leading Methods

We compare our method with multiple MLLMs, including two types of state-of-the-art models: 1) General MLLMs. We include LLaVA [2], InstructBLIP [4], LLaVA-V1.5 [3], and Qwen-VL-Chat [6] as high-performing, open-sourced general models. These models are trained on extensive datasets and demonstrate promising results across various tasks. 2) RLHF-based MLLMs. Our comparisons also extend to RLHF models such as LLaVA-RLHF [11], RLHF-V [12], POVID [14], SILKIE [34], and FGAIF [13]. Specifically, LLaVA-RLHF employs the PPO algorithm on 10k human-preference data and 72k factually augmented data for reward and policy models, respectively. RLHF-V utilizes 1.4k human-annotated, fine-grained preference data to optimize the policy model using the DPO algorithm. Both [13] and [34] apply the DPO algorithm to align MLLMs with GPT-4V preferences. POVID [14] generates hallucination examples through two strategies and also uses the DPO algorithm.

The quantitative results are shown in Table 1 and 2. We observe that our AMP surpasses all general MLLMs across all benchmarks, highlighting the benefits of further fine-tuning with the preference dataset. Besides, our method also achieves state-of-the-art performance among RLHF-based methods, which comes from two aspects. First, our MDPO algorithm facilitates multi-level preference learning, which enables the MLLM to discern semantic granularity among different responses. Second, the accurate ranking of our human-free preference dataset ensures reliable guidance for the MLLM, leading to more promising performance. We also provide some qualitative case studies in Fig 3. For more cases, please refer to Appendix A.5.

Table 2: Comparisons on the POPE benchmark. * indicates evaluations using the official model.

| Methods | Adversarial | | Popular | | Random | | Overall | |
| --- | --- | --- | --- | --- | --- | --- | --- | --- |
| | F1↑ | Acc.↑ | F1↑ | Acc.↑ | F1↑ | Acc.↑ | F1↑ | Yes |
| LLaVA$_{13B}$ [2] | 74.4 | 67.2 | 78.2 | 73.6 | 78.8 | 73.7 | 77.1 | 73.7 |
| InstructBLIP$_{7B}$ [4] | 70.4 | 65.2 | 80.2 | 79.7 | 89.3 | 88.6 | 80.0 | 59.0 |
| DeepSeek-VL [44] | 72.2 | 65.4 | 71.3 | 63.8 | 76.4 | 71.5 | 71.7 | 73.3 |
| Qwen-VL-Chat [6] | 80.7 | 83.2 | 81.6 | 84.2 | 82.1 | 84.2 | 81.5 | 36.7 |
| MiniCPM-V [45] | 83.5 | 83.4 | 86.2 | 86.5 | 88.9 | 89.2 | 86.2 | 47.8 |
| LLaVA-V1.5$_{7B}$ [3] | 84.5 | 85.5 | 86.0 | 87.1 | 87.2 | 88.0 | 85.9 | 42.2 |
| LLaVA-V1.5$_{13B}$ [3] | 84.5 | 85.5 | 86.3 | 87.4 | 87.1 | 88.0 | 86.0 | 42.2 |
| LLaVA-V1.6$_{7B}$ [3] | **85.2** | **86.4** | 86.4 | 87.6 | 87.6 | 88.5 | 86.4 | 41.5 |
| LLaVA-V1.6$_{13B}$ [3] | **85.2** | **86.4** | 86.4 | 87.7 | 87.2 | 88.2 | 86.3 | 41.0 |
| LLaVA-RLHF$_{7B}$ [11] | 79.5 | 80.7 | 81.8 | 83.3 | 83.3 | 84.8 | 81.5 | 41.8 |
| LLaVA-RLHF$_{13B}$ [11] | 80.5 | 82.3 | 81.8 | 83.9 | 83.5 | 85.2 | 81.9 | 39.0 |
| RLHF-V [12] | 83.6 | 84.6 | 85.3 | 86.4 | 87.2 | 88.1 | 85.4 | 42.7 |
| POVID* [14] | 84.0 | 84.7 | 85.8 | 86.8 | 87.7 | 88.5 | 85.8 | 43.6 |
| SILKIE [34] | 80.3 | 83.0 | 81.3 | 84.0 | 81.6 | 83.9 | 81.1 | 36.1 |
| FGAIF [13] | 79.9 | 79.6 | 83.7 | 84.0 | 86.7 | 87.0 | 83.4 | 48.3 |
| AMP-MEG$_{7B}$ | 83.4 | 83.1 | 87.7 | **88.2** | 89.6 | 89.9 | 86.9 | 48.0 |
| AMP-MEG$_{13B}$ | 83.4 | 82.8 | **88.0** | **88.2** | **90.3** | **90.4** | **87.2** | 49.8 |
| AMP-IG$_{7B}$ | 82.3 | 82.5 | 87.0 | 87.8 | 87.7 | 88.3 | 85.7 | 45.1 |
| AMP-IG$_{13B}$ | 83.0 | 82.7 | 86.0 | 86.3 | 89.6 | 90.0 | 86.2 | 46.3 |

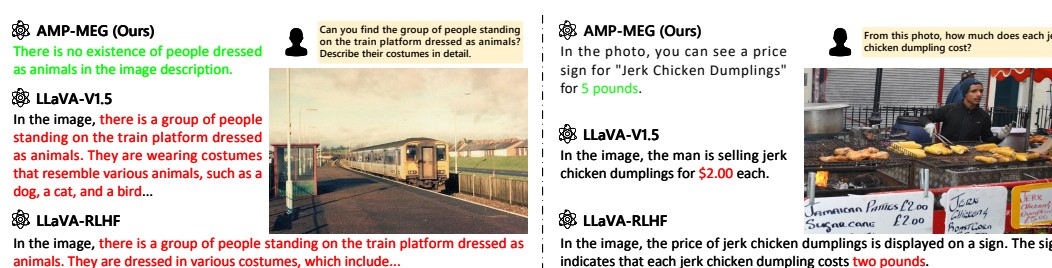

Figure 3: Case studies including our AMP-MEG, LLaVA-V1.5 [3], and LLaVA-RLHF [11]. Hallucinations, correct responses are highlighted in different colors. Please zoom in for the best view.

Table 3: Study on preference quantity.

| Settings | MMHal-Bench | | MRHal-Bench | | LLaVA-Bench | | |
| --- | --- | --- | --- | --- | --- | --- | --- |
| | Score↑ | Hal.↓ | Score (c/m)↑ | Hal.↓ | Conv.↑ | Detail↑ | Comp.↑ |
| 2-level preference | 2.69 | 0.47 | 3.71 / 3.72 | 0.27 / 0.22 | 81.5 | 74.7 | 83.5 |
| 3-level preference | 2.88 | 0.42 | 3.83 / 3.87 | 0.24 / 0.18 | 84.1 | 84.6 | 94.1 |
| 4-level preference | **3.17** | **0.35** | **4.07 / 4.06** | **0.20 / 0.15** | **89.7** | **89.1** | **98.8** |
| 5-level preference | 2.96 | 0.41 | 3.93 / 3.95 | 0.22 / 0.17 | 88.5 | 84.8 | 92.9 |

## 4.4 Ablation Studies

**Impact of Preference Quantity.** We explore the effects of varying the number of preferences from 2 to 5, with detailed implementation found in Appendix A.6. As indicated in Table 3, a 4-level preference is identified as the optimal setting. We hypothesize that the diminished performance observed with a 5-level preference dataset may be due to increased hidden noise. Unless stated otherwise, all subsequent experiments are conducted on an MLLM using the Vicuna-7B, trained on the 4-level preference dataset produced through Multi-size Expert Generation.

Table 4: Impact of the gap between adjacent levels and cross-level comparison. Preferences are ranked from most superior to most inferior in the following order: S, A, B, C.

| Settings | MMHal-Bench | | MRHal-Bench | | LLaVA-Bench | | |
|---|---|---|---|---|---|---|---|
| | Score↑ | Hal.↓ | Score (c/m)↑ | Hal.↓ | Conv.↑ | Detail↑ | Comp.↑ |
| S>B | 2.50 | 0.50 | 3.56 / 3.57 | 0.28 / 0.28 | 79.7 | 71.5 | 80.3 |
| S>A | 2.61 | 0.51 | 3.63 / 3.62 | 0.27 / 0.25 | 82.8 | 74.9 | 84.1 |
| S>A & A>B | 2.68 | 0.43 | 3.69 / 3.71 | 0.29 / 0.22 | 83.0 | 79.6 | 87.0 |
|   +Cross-level Comparison | 2.79 | 0.44 | 3.75 / 3.73 | 0.27 / 0.22 | 87.7 | 78.6 | 90.1 |
| S>A & A>B & B>C | 2.85 | 0.40 | 3.86 / 3.90 | 0.24 / 0.17 | **90.2** | 81.3 | 92.4 |
|   +Cross-level Comparison | **3.17** | **0.35** | **4.07 / 4.06** | **0.20 / 0.15** | 89.7 | **89.1** | **98.8** |
| A>C | 2.33 | 0.57 | 3.37 / 3.37 | 0.34 / 0.34 | 75.7 | 70.4 | 80.1 |
| B>C | 2.45 | 0.51 | 3.50 / 3.50 | 0.31 / 0.27 | 77.3 | 72.2 | 81.6 |

Table 5: Ablations on the human-free multi-level preference dataset using different annotations, including AI (*i.e.*, GPT-4V), Auto-check, and initial annotations from MEG and IG.

| Dataset | Preference Annotation | MMHal-Bench | | MRHal-Bench | | LLaVA-Bench | | |
|---|---|---|---|---|---|---|---|---|
| | | Score↑ | Hal.↓ | Score (c/m)↑ | Hal.↓ | Conv.↑ | Detail↑ | Comp.↑ |
| AMP-MEG | AI | 2.87 | 0.44 | 3.87 / 3.88 | 0.25 / 0.21 | 92.3 | 79.6 | 89.9 |
| AMP-MEG | Auto Check | **3.17** | **0.35** | **4.07 / 4.06** | **0.20** / 0.15 | 89.7 | **89.1** | 98.8 |
| AMP-IG | Auto Check | 3.12 | 0.41 | 4.02 / 4.04 | 0.22 / **0.13** | 90.2 | 85.9 | **99.8** |
| AMP-MEG | Initial | 2.79 | 0.49 | 3.69 / 3.71 | 0.29 / 0.22 | 87.1 | 80.1 | 81.7 |
| AMP-IG | Initial | 2.80 | 0.48 | 3.75 / 3.77 | 0.26 / 0.21 | 89.2 | 78.3 | 81.9 |

**Impact of Gap between Adjacent Levels.** The effectiveness of multi-level preference learning is partly attributed to reducing the gaps between adjacent levels. As shown in Table 4, we reduce the gap by "S>B ⇒S>A" and "A>C ⇒B>C", both of which result in performance enhancements.

**Impact of Including More Comparisons.** We further introduce more inferior responses, *i.e.*, 'Response B' and 'Response C', by "S>A ⇒ S>A & S>B" and "S>A & S>B ⇒ S>A & A>B & B>C". The improvements depicted in Table 4 verify that inferior responses are also beneficial for preference learning. To provide more comparisons between the best response and hallucination examples, we devise cross-level comparisons based on settings "S>A & S>B" and "S>A & A>B & B>C". As illustrated in Table 4, this strategy brings extra performance improvement across multiple benchmarks, indicating the necessity of cross-level comparisons.

**Comparisons with AI-Annotated Preference.** Similar to reinforcement learning methods using AI feedback [46], we use GPT-4V [15] to directly rank the responses generated by MEG based on their visual faithfulness and helpfulness. Table 5 illustrates that training with our preference dataset yields more effective results compared to the AI-annotated preference dataset. This suggests that our human-free multi-level preference dataset contains less noise. Furthermore, the performance of the MLLM significantly decreases in the absence of our Auto-check mechanism, highlighting its crucial role in accurately refining the ranking of the multi-level preference dataset.

### 4.5 Comparisons with other Rank-based Preference Alignment Approaches.

We make some empirical comparisons by replacing our MDPO with the learning objectives of [47] and [48]. As reported in Table 6, our MDPO surpasses these two learning objectives on all hallu-cination benchmarks, *e.g.*, 3.01 -> 3.17 on MMHal-Bench. The superiority of our MDPO comes from two aspects. First, our MDPO mitigates the challenge of distinguishing micro hallucinations in responses. Taking 3-level preference as an example, the comparisons of other methods are 'A>BC, B>C', while comparisons made by our AMP are 'A>B, A>C, B>C'. More specifically, our AMP splits 'A>BC' into 'A>B, A>C', which enables MLLMs to perceive the subtle differences between different responses. Second, our penalty term explicitly increases the probability of MLLMs generating good answers, ensuring the stability of the training process.

We also conduct some experiments about perturbation-based (PB) methods. Our implementation details are as follows. We randomly change the noun, adjective, preposition, and numeral. We obtain

Table 6: Performance on three hallucination benchmarks across other loss functions (#1, #2), MLLMs from different families (#3, DF), perturbation-based methods (#4, PB).

| Settings | MMHal-Bench | | MRHal-Bench | | LLaVA-Bench | | |
| --- | --- | --- | --- | --- | --- | --- | --- |
| | Score↑ | Hal.↓ | Score↑ | Hal.↓ | Conv.↑ | Detail↑ | Comp.↑ |
| r. [47] | 2.96 | 0.41 | 3.85 / 3.82 | 0.26 / 0.23 | 84.1 | 81.7 | 88.2 |
| r. [48] | 3.01 | 0.38 | 3.95 / 3.91 | 0.24 / 0.19 | 86.2 | 84.3 | 91.9 |
| PB | 2.83 | 0.46 | 3.61 / 3.52 | 0.33 / 0.35 | 78.4 | 75.1 | 81.3 |
| MDPO | **3.17** | **0.35** | **4.07 / 4.06** | **0.20 / 0.15** | **89.7** | **89.1** | **98.8** |

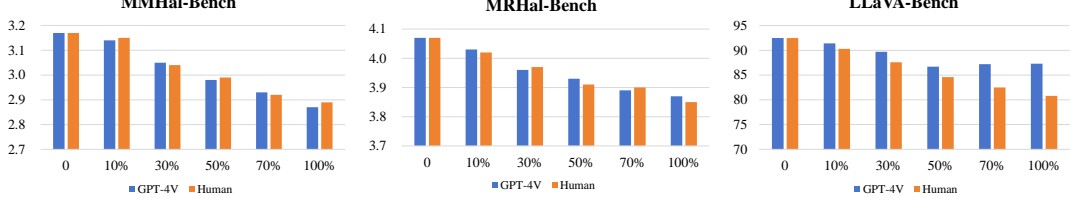

Figure 4: Performance on three hallucination benchmarks across different proportions of GPT-4V/human annotations.

answers of varying quality by controlling the proportion of perturbations (10%, 30%, 50% based on 4-level preference setting). As shown in Table 6, perturbation-based methods is still inferior to our MDPO. We infer the hallucination pattern generated by random perturbation is different from the real MLLM and is thus not informative enough for preference learning.

### 4.6 Evaluation of the Automated Multi-level Preference Dataset

We estimate the inconsistency rate of our AMP dataset to be 2.25% (through manual evaluation on 2000 random samples). The 2.25% inconsistency rate is significantly lower than the human (14.40% inconsistency) and GPT-4V (11.95% inconsistency) annotations.

Moreover, we conduct another experiment to validate the superiority of our AMP dataset. We mix the AMP and human/GPT-4V data for training the model. Fig. 4 shows that as the proportion of human/GPT-4V annotated data increases, the performance of MLLMs decreases accordingly.

## 5 Limitations

Our AMP framework offers more effective preference learning from the human-free multi-level preference dataset. However, several challenges remain: 1) The quality of standard responses limits the performance of the optimized MLLM. A portion of the standard responses in our dataset comes from superior responses generated by language models, potentially containing imperceptible hallucinations. Besides, the standard response is less helpful despite its high faithfulness, further restricting the performance. 2) Although our AMP successfully reduces hallucinations and promotes the truthfulness of MLLMs, the essence of preference learning is pushing the model to bias the preference dataset, causing a decrease in generalization ability. Therefore, finding a balance between preference learning and maintaining the capabilities of MLLMs is yet to be explored.

## 6 Conclusions

In this paper, we introduce the Automated Multi-level Preference (AMP) framework, achieving promising performance on several hallucination benchmarks, which benefits from the reduction of gaps in adjacent levels and the introduction of cross-level comparison. To enable the AMP framework, we propose a multi-level preference dataset generation pipeline, aiming to construct a high-quality preference dataset automatically. Furthermore, we design the Multi-level Direct Preference Optimization algorithm, which furnishes a novel learning objective to ensure robust and efficient preference learning. Lastly, we conduct the first hallucination benchmark in multi-round dialogues and devise the relevant metrics, which may stimulate future research.

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

# A   Appendix

## A.1   Omitted Nouns in Auto-check Mechanism

Similar to [5], we exclude some abstract nouns, *e.g.*, "time", "effect", *etc.* Besides, some high-frequency but unnecessary nouns, such as "image", "photo", *etc*, are also deprecated. The complete list is depicted in Fig. 5.

> **Unnecessary Nouns**: 'photo', 'image', 'photograph', 'picture', 'figure', 'painting'.
>
> **Abstract Nouns**: 'harmony', 'beauty', 'feel', 'concentration', 'life', 'matter', 'feature', 'effect', 'skill', 'detail', 'emotion', 'nature', 'time', 'background', 'placement', 'perspective', 'universality', 'attention', 'features', 'essence', 'peace', 'composition', 'element', 'foreground', 'appreciation', 'atmosphere', 'emotions', 'artwork', 'subject'.

Figure 5: Omitted nouns in the auto-check mechanism.

## A.2   Annotated Process of Multi-round Dialogues in Training Dataset

We introduce 2k multi-round dialogues in our training dataset. To get diverse questions, we use GPT-4V [15] to generate questions and corresponding responses, where the prompt is as Fig. 6.

However, the responses generated by GPT-4V still contain hallucinations. To get the high-quality response, we further employ Qwen-VL-Chat [6], LLaVA-v1.5 13B [3], LLaVA-RLHF 13B [11], RLHF-V [12] to provide three extra responses. Thus, together with the response of GPT-4V, we get 5 responses in total. Then, we send these responses to human annotators and ask them to find the optimal response. If the optimal response still exhibits hallucinations, this image-text pair will be deprecated. Through these steps, we get the final 2k high-quality multi-round dialogues.

## A.3   Details of Human-free Multi-level Preference Dataset Generation

### A.3.1   Multi-size Expert Dataset Generation

To make the language style more consistent, we use models from the same family, including LLaVA-2B, LLaVA-7B, LLaVA-13B, and LLaVA-34B. We leverage the greedy decoding strategy with specific parameters: beams (4), temperature (0.7), repetition penalty (1.1), and max tokens (512).

### A.3.2   Incremental Generation

In the Incremental Generation Generation, we obtain MLLMs with varying capabilities by using train datasets of different sizes. Specifically, we use the 7B version of LLaVA-V1.5 as the pre-trained model and follow the training detail of [3] and further fine-tune it on 30k/60k/90k high-quality image-text pairs. The whole dataset contains 10k ShareGPT4V [16], 20k Flickr30k [49], 30k VQAv2 [50], and 30k LRV [26]. Finally, we get 5 different responses, including the Ground Truth (GT), responses generated by 3 fine-tuned MLLMs and LLaVA-V1.5.

## A.4   Multi-round Dialogue Hallucination Benchmark (MRHal-Bench)

To evaluate the hallucinations in multi-round dialogue, we build a Multi-round Dialogue Hallucination Benchmark, simplified by "MRHal-Bench". Specifically, MRHal-Bench contains 105 multi-round dialogues, where the length of rounds ranges from 2 to 5, with an average length of 2.99. The questions in MRHal involve five categories where MLLMs tend to generate hallucinations:

- Attribute: Visual characteristics of objects, including color, shape, state, type, *etc*.
- Description: Detailed descriptions of objects, behaviors, environments, background, foreground, *etc*.

You are an image content annotation expert, and you are seeing a single image. Design a multi-round conversation between you and a person asking about this photo.

The conversation format is
User: ********
GPT: ********
where the user is the role of the person, and GPT is the role of you.

The answers should be in a tone that a visual AI assistant is seeing the image and answering the question. Ask diverse questions and give corresponding answers. Only include questions that have definite answers:
(1) Ask the questions that can be answered confidently. Do not ask ambiguous questions. Do not give any ambiguous answers.
(2) Your answers may involve some facts about the image, such as news, social, science, etc.

First, your questions may include the characteristics of all the objects or elements in the image, their type, color, and style, the number of objects, the movement of the characters, the precise location, text, etc. Also include complex questions that are relevant to the content in the image, for example, asking about background knowledge of the objects in the image, asking to discuss events happening in the image, etc. Again, do not ask about uncertain details, do not imagine anything. Provide detailed answers when answering complex questions. For example, give detailed examples or reasoning steps to make the content more convincing and well-organized. You can include multiple paragraphs if necessary. Your output should only contain the conversations without any other things.

Figure 6: The text prompt for GPT-4V in the annotated process of multi-round dialogue.

- Existence: Questions with absolute answers, *e.g.*, yes/no.
- Counting: The number of specific objects.
- Reasoning: Questions that require the model to integrate analysis based on image content or other knowledge to generate final responses.
- Spatial Relation: The relative or absolute spatial relationships of objects in given images.

To obtain a fair, objective, and trustable evaluation result, we employ GPT-4 via a meticulous template. Since GPT-4V also suffers from hallucinations, we use GPT-4 API and replace visual contents with category names and a standard human-generated answer, which is similar with [11]. The template will be released in our code.

We design two types of metrics for evaluating MRHal-Bench, *i.e.*, cumulative and mean metrics (denoted by c/m in Section 4). The cumulative ($Metric_\mathrm{c}$) and mean scores ($Metric_\mathrm{m}$) are calculated by,

$$Metric_\mathrm{c} = \sum_{i=1}^{105} \left[ \mathrm{Sum}\left(Metric_i\right)\right] / \sum_{i=1}^{105} \left[ \mathrm{Len}\left(Metric_i\right)\right],$$

$$Metric_\mathrm{m} = \left\{ \sum_{i=1}^{105} \left[ \mathrm{Sum}\left(Metric_i\right) / \mathrm{Len}\left(Metric_i\right)\right]\right\} / 105, \tag{8}$$

where $Metric_i \in \mathbb{R}^j$ symbolizes the Metric of $i$-th dialogue with $j$ rounds.

MRHal-Bench will be available at this link.

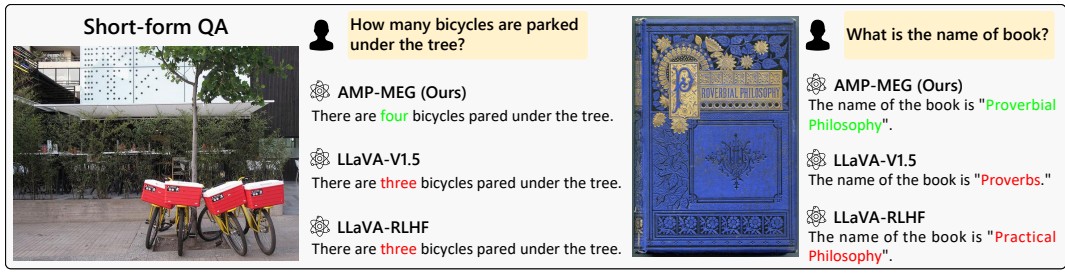

**Short-form QA**

👤 How many bicycles are parked under the tree?

⚛ **AMP-MEG (Ours)**
There are four bicycles pared under the tree.

⚛ **LLaVA-V1.5**
There are three bicycles pared under the tree.

⚛ **LLaVA-RLHF**
There are three bicycles pared under the tree.

👤 What is the name of book?

⚛ **AMP-MEG (Ours)**
The name of the book is "Proverbial Philosophy".

⚛ **LLaVA-V1.5**
The name of the book is "Proverbs."

⚛ **LLaVA-RLHF**
The name of the book is "Practical Philosophy".

**Long-form QA**

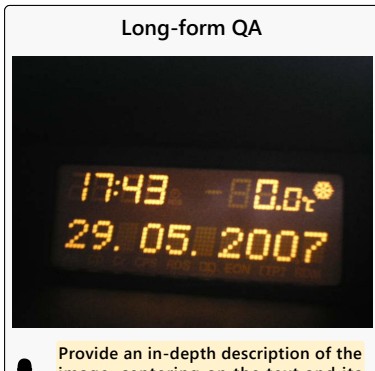

👤 Provide an in-depth description of the image, centering on the text and its context.

⚛ **AMP-MEG (Ours)**
The image shows the inside of a car's dashboard, displaying a digital temperature reading of "0°C" and the time "5:43 pm". The car is likely in Europe, as the temperature is displayed in Celsius. There is also an image of a snowflake symbol visible, indicating that it might be cold enough for snow to form. The background of the image is not in clear view, but it can be inferred that the photo was taken in 2007.

⚛ **LLaVA-V1.5**
The image features a digital display screen, likely from a car, showing the current time and temperature. The time is displayed in the upper left corner of the screen, while the temperature is shown in the lower right corner. The temperature is -8 degrees Celsius, indicating a cold day. The display screen is illuminated, making the numbers and text visible and easy to read.

⚛ **LLaVA-RLHF**
The image features a car dashboard with a digital clock and temperature display. The clock is set to 17:25, indicating the time of day. The temperature display shows a chilly -8 degrees Celsius, suggesting that it is a cold day outside. The car's dashboard is equipped with a variety of digital displays, including a speedometer, fuel gauge, and other essential information for the driver. The presence of these displays indicates that the car is well-equipped and designed to provide the driver with all necessary information to ensure a safe and comfortable journey...

**Existence Hallucinations**

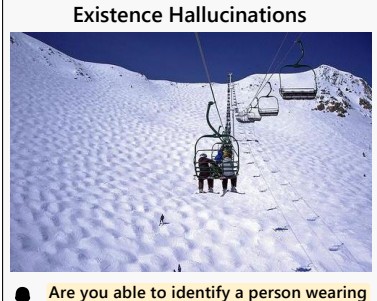

👤 Are you able to identify a person wearing a bright yellow helmet in the image?

⚛ **AMP-MEG (Ours)**
There is no existence of a bright yellow helmet in the image.

⚛ **LLaVA-V1.5**
Yes, there is a person wearing a bright yellow helmet in the image.

⚛ **LLaVA-RLHF**
Yes, I can see a person wearing a bright yellow helmet in the image. This person is likely a skier or snowboarder, as they are riding a ski lift up a snowy mountain. The presence of the yellow helmet suggests that the person is taking safety precautions while engaging in winter sports activities.
Wearing a helmet is essential for protecting the head from potential injuries during skiing or snowboarding. It can help prevent serious head injuries, as well as provide protection from cold weather conditions. The bright color of the helmet also makes it easier for others to spot the person on the slopes, which can be helpful in case of emergencies or when visibility is limited.

**Image Caption**

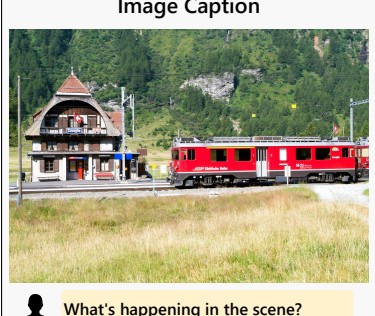

👤 What's happening in the scene?

⚛ **AMP-MEG (Ours)**
The image captures a picturesque scene of rural life in the Swiss Alps. A red and white electric train, bearing the logo of the Swiss Federal Railways, is seen traveling along a track that runs adjacent to a building that resembles a mountain station. The building, with its peaked roof and white walls...

⚛ **LLaVA-V1.5**
The image features a red train traveling down the tracks, passing by a small building, which could be a house or a station. The train appears to be a passenger train, as it is pulling into the station. There are several people visible in the scene, with some standing near the train and others scattered around the area. They may be waiting for the train to come to a complete stop or simply observing its journey...

⚛ **LLaVA-RLHF**
The image features a red train traveling down the tracks, passing by a small building and a mountain. The train is positioned in the middle of the scene, with the building located on the left side and the mountain in the background. There are several people visible in the scene, with some standing near the train and others scattered around the area. A car can also be seen parked near the train, possibly belonging to one of the people in the scene.

Figure 7: Case studies in terms of short-form Question Answering (QA), long-form QA, Existence Hallucinations, and Image Caption, including our AMP-MEG, LLaVA-V1.5 [3], and LLaVA-RLHF [11]. Hallucinations, correct responses are highlighted in different colors.

## A.5 Case Studies

In Fig. 7, we present some examples of our AMP alongside other MLLMs, *i.e.*, LLaVA-V1.5 [3], and LLaVA-RLHF [11] for an intuitive comparison. We focus on four typical scenarios. 1) Short-form

Question Answering (QA). Our AMP generally provides accurate responses, such as the counting (*e.g.*, "four bicycles") and character (*e.g.*, "Proverbial Philosophy"). 2) Long-form QA. Our AMP outperforms other MLLMs in terms of helpfulness and faithfulness. Specifically, our AMP accurately interprets all the valuable information in the given image, including the time, date, and temperature. In contrast, other MLLMs make a wrong judgment about or neglect the information in this image. 3) Existence Hallucinations. Compared with responses from other MLLMs, our AMP is not misdirected by "identify a person wearing a bright yellow helmet" and predicts the non-existence of this person. 4) Image Caption. For the detailed caption, our AMP captures all the significant visual components correctly and infers the location ("Swiss Alps") from the Swiss national flag and mountains. However, other MLLMs overlook this flag and generate some hallucinations about people and cars. These qualitative results verify the superiority of our AMP framework.

### A.6 Implementation Details of Optimal-Level Experiments

In Section 4.4, we report the performance of $K$-level preferences, where $K$ ranges from 2 to 5. When $K$ is equal to 4, the refined MLLM gets the best performance. However, the performance of optimized MLLM varies from different model pools. Take $K = 3$ as the example, the model pool may be 'Response S&34B&13B', 'Response S&13B&7B', *etc*. The performance of optimized MLLM varies from different model pools. Therefore, we only report the optimal performance under each level, where the details of model pools are reported in Table 7.

Table 7: The model pools for each level preference.

| Settings | GT | LLaVA-34B | LLaVA-13B | LLaVA-7B | LLaVA-2B |
|---|---|---|---|---|---|
| 2-level preference | ✓ | ✓ | | | |
| 3-level preference | ✓ | ✓ | ✓ | | |
| 4-level preference | ✓ | | ✓ | ✓ | ✓ |
| 5-level preference | ✓ | ✓ | ✓ | ✓ | ✓ |

