# OpenReview forum: "Automated Multi-level Preference for MLLMs"
_NeurIPS.cc/2024/Conference — NeurIPS 2024 poster_

### Official Review · Reviewer_xo7T · 2024-07-07

**Soundness:** 4
**Presentation:** 3
**Contribution:** 4
**Rating:** 6
**Confidence:** 4

**Summary:**

This paper presents the Automated Multi-level Preference (AMP) framework for improving MLLMs by addressing hallucination issues. The framework introduces a multi-level preference system for RLHF, aiming to enhance the learning process by providing more granular feedback.

**Strengths:**

- The introduction of multi-level preferences rather than binary ones narrows the gap between adjacent levels, enabling MLLMs to discern subtle differences and integrate cross-level comparisons.
- The automated pipeline for generating high-quality multi-level preference datasets without human annotators is a significant contribution, potentially reducing bias and noise while saving time and resources.
- Extensive experiments across multiple benchmarks demonstrate the effectiveness of the proposed method.

**Weaknesses:**

- The contribution of the paper heavily relies on the preference fine-tuning algorithm, showing limited innovation beyond this aspect.
- The method does not demonstrate significant improvements on the LLaVA-Bench benchmark.
- The method's performance on the adversarial tasks of the POPE benchmark is moderate, suggesting a need to reconsider the impact of MDPO on model robustness and how to balance performance and robustness.

**Questions:**

See weaknesses.

---

> ### Author Rebuttal · Authors · 2024-08-07
>
> **Q1.** The contribution of the paper heavily relies on the preference fine-tuning algorithm, showing limited innovation beyond this aspect.
>
> **A1.** In this paper, we introduce the Automated Multi-level Preference (AMP) framework, which involves generating high-quality multi-level preference datasets and the effective learning objective for MDPO. These novel designs ensure AMP achieve promising performance on several hallucination benchmarks. Moreover, we conduct the first hallucination benchmark in multi-round dialogues and devise the relevant metrics, which may stimulate future research.
>
> ---
> **Q2.** The method does not demonstrate significant improvements on the LLaVA-Bench benchmark.
>
> **A2.** Despite a slight performance degradation on the LLava-Bench, our AMP shows significant improvements compared to both general MLLMs and RLHF methods on other benchmarks, $e.g.$, 3.09 (FGAIF) -> 3.23 on MMHal-Bench, 3.71/3.70 (SILKIE) -> 4.21/4.21 on MRHal-Bench, and 85.8 (POVID) -> 87.2 on POPE.
>
> ---
> **Q3.** The method's performance on the adversarial tasks of the POPE benchmark is moderate, suggesting a need to reconsider the impact of MDPO on model robustness and how to balance performance and robustness.
>
> **A3.** The principle of RLHF is explicitly enlarging the probability of superior responses while decreasing the probability of inferior responses. Compared to the conventional autoregressive function, loss in RLHF ensures MLLMs to fit the characteristics of the dataset at the sacrifice of generation ability/robustness. In the future, we will consider integrating the advantages of autoregressive function to maintain the robustness of MLLMs.

---

> > ### Comment · Reviewer_xo7T · 2024-08-08
> > **Official Comment by Reviewer xo7T**
> >
> > Thank you for the rebuttal, which addressed some of my concerns. I have increased my score and am looking forward to reading your revision in future venue.

---

### Official Review · Reviewer_iDRq · 2024-07-10

**Soundness:** 3
**Presentation:** 3
**Contribution:** 3
**Rating:** 5
**Confidence:** 5

**Summary:**

In this paper, the authors develop an automated dataset generation pipeline capable of producing multi-level preference datasets without the need for human annotators. This paper introduces a novel multi-round dialogues hallucination benchmark, MRHal-Bench. Additionally, the authors design the Multi-level Direct Preference Optimization (MDPO) algorithm, which employs a specifically crafted learning objective to facilitate multi-level preference learning. Extensive experiments conducted on both the hallucination benchmark and a general benchmark demonstrate the effectiveness of this method.

**Strengths:**

1. To make the labeling of multi-level preference datasets cost-effective and efficient, this paper proposes an automated dataset generation pipeline capable of producing high-quality preference datasets.

2. To narrow the gap between two preference samples in DPO and make the model more easily distinguish the differences between preference data, this paper proposes a multi-level DPO algorithm that use multi-level preference data to provide a broader range of comparisons with hallucination examples.

**Weaknesses:**

1. It is recommended to provide more quantitative information on the preference dataset generated by the automated dataset generation pipeline. For instance, the authors could use a subset of the dataset to demonstrate the similarity results compared to human annotators.
2. In this paper, the authors conduct experiments on three hallucination benchmarks and only one general benchmark. To verify the more general applicability of the method, additional experiments are needed on general benchmarks such as TextVQA, GQA, and IconQA.
3. In Table 1, the authors compare several MLLMs and RLHF-based MLLMs across MMHal-Bench, MRHal-Bench and LLaVA-Bench. However, the baseline model should be more up-to-date. Could you compare it with more current models such as LLaVA-v1.6, DeepSeek-VL, or MiniCPM-V?

**Questions:**

1. Assume we have 3 preference samples: A, B, C. Using the MDPO algorithm, we need to calculate the loss for AB, AC, and BC and then update the parameters. However, why do we need to calculate the loss for BC? Sample B may contain hallucinations; does this affect the model's learning of correct preferences?
2. This paper does not enhance the visual capabilities of the model. However, in the case study, several OCR tasks and the AMP-MEG model can successfully recognize. Can the authors explain why MDPO algorithm can improve this aspect of ability?

**Limitations:**

Yes.

---

> ### Author Rebuttal · Authors · 2024-08-07
>
> **Q1.** It is recommended to provide more quantitative information on the preference dataset generated by the automated dataset generation pipeline. For instance, the authors could use a subset of the dataset to demonstrate the similarity results compared to human annotators.
>
> **A1.** Thanks for raising this good discussion. We provide more intrinsic evaluation for the automated multi-level preference dataset and the auto-check mechanism.
>
> **Evaluation of the automated multi-level preference dataset.** During rebuttal, we estimated the inconsistency rate of our AMP dataset to be 2.25% (through manual evaluation on 2000 random samples). The 2.25\% inconsistency rate is significantly lower than the human (14.40% inconsistency) and GPT-4V (11.95% inconsistency) annotations. Below is an example of an inconsistent case in our AMP dataset, where $N_i$ denotes the $i$-th noun chunks:
>
> >**Standard Response**: A little girl ($N_1$) with a purple jacket ($N_2$) is flying a kite ($N_3$).
> >
> >**Response A**: A little girl ($N_1$) dressed in purple jacket ($N_2$) is flying a kite ($N_3$) on the lawn, surrounded by *many people* (other information, including some *hallucinations* denoted by *italic*).
> >
> >**Response B**: A young girl ($N_1$) dressed in purple clothes ($N_4$, similar to $N_2$ but not exactly the same) is flying a kite ($N_3$).
>
> Response A accurately predicts these noun chunks and thus gets a higher score, leading to A>B. However, the actual ranking is B>A because A has some hallucinations.
>
> Moreover, we conduct another experiment to validate the superiority of our AMP dataset. We mix the AMP and human/GPT-4V data for training the model. Table 1 in the PDF document shows that as the proportion of human/GPT-4V annotated data increases, the hallucination rate rises accordingly.
>
> ---
>
> **Q2.** In this paper, the authors conduct experiments on three hallucination benchmarks and only one general benchmark. To verify the more general applicability of the method, additional experiments are needed on general benchmarks such as TextVQA, GQA, and IconQA.
>
> **A2.** Compared to the baseline, metrics of the MLLM fine-tuned with our MDPO are improved by 3.3 (58.2 -> 61.5) and 1.8 (62.0 -> 63.8) on two general benchmarks, TextVQA and GQA, respectively.
>
> ---
>
> **Q3.** In Table 1, the authors compare several MLLMs and RLHF-based MLLMs across MMHal-Bench, MRHal-Bench and LLaVA-Bench. However, the baseline model should be more up-to-date. Could you compare it with more current models such as LLaVA-v1.6, DeepSeek-VL, or MiniCPM-V?
>
> **A3.** We present the performance LLaVA-v1.6, DeepSeek-VL, and MiniCPM-V in Table 3 (PDF document).
> As shown in Table 3, our method significantly exceeds general MLLMs on hallucination benchmarks, $e.g.$, +0.13 (LLaVA-V1.6) on MMHal-Bench, +0.27/+0.28 (LLaVA-V1.6) on MRHal-Bench, verifying the effectiveness of preference learning and our AMP pipeline.
>
> ---
>
> **Q4.** Assume we have 3 preference samples: A, B, C. Using the MDPO algorithm, we need to calculate the loss for AB, AC, and BC and then update the parameters. However, why do we need to calculate the loss for BC? Sample B may contain hallucinations; does this affect the model's learning of correct preferences?
>
> **A4.** First, loss for BC provides more comparisons among hallucination examples, resulting in a better performance. Second, to alleviate the adverse effects caused by hallucinations in B, we only introduce the penalty term for AB and AC and exclude the penalty term for BC.
>
> We also conduct another experiment for loss BC. As reported in Table 4 (PDF document), the loss for BC brings extra performance improvement.
>
> ---
>
> **Q5.** This paper does not enhance the visual capabilities of the model. However, in the case study, several OCR tasks and the AMP-MEG model can successfully recognize. Can the authors explain why MDPO algorithm can improve this aspect of ability?
>
> **A5.** Improvement on OCR is because our AMP dataset actually contains some preference pairs that are centric to OCR task. Below is an example:
>
> >**Prompt**: What does the sign read?
> >
> >**Correct Response**: NO TIPPING THIS AREA IS MONITORED BY 24 HOUR RECORDED CCTV.
> >
> >**Response A**: NO TIPPING AREA IS MONITORED BY 24-HOUR RECORDED CCTV.
> >
> >**Response B**: NO TIPPING MONITOR 24 HOUR RECORDED CCTV.
> >
> >**Response C**: NO TIPPING.
>
> Preference learning framework can increase the probability of correct responses while decreasing the probability of other responses, which enhances the OCR ability.

---

> > ### Comment · Reviewer_iDRq · 2024-08-11
> >
> > The author has addressed most of the concerns. The reviewer maintains the initial rating and is inclined towards acceptance of the paper.

---

### Official Review · Reviewer_2sCy · 2024-07-29

**Soundness:** 2
**Presentation:** 3
**Contribution:** 2
**Rating:** 6
**Confidence:** 4

**Summary:**

This work aims to mitigate hallucinations in Multimodal Large Language Models through preference optimization. Motivated by two limitations of binary preferences widely used in existing work, authors proposed a multi-level preference framework. The framework consists of 1) an automated dataset generation pipeline that converts each image-text pair into an image with multiple text descriptions from superior to inferior quality 2) a Multi-level Direct Preference Optimization algorithm that enumerates over all preference pairs with the standard DPO objective. Additionally, authors introduce a new hallucination benchmark, MRHal-Bench. The proposed framework has been evaluated on three benchmarks: MMHal-Bench, LLaVA-Bench, and MRHal-Bench against 5 base models and 5 preference fine-tuned model. The proposed framework achieves best state-of-the-art on MMHal-Bench and MRHal-Bench, although only improved over the second best FGAIF by a small margin. Authors also include comprehensive ablation studies on the effects of multi-level preference.

**Strengths:**

* The application of multi-level preference alignment to the problem of mitigating hallucination in multimodal LLMs is novel.
* Conduct a comprehensive comparison with existing preference fine-tuned multimodal LLMs and baselines on three benchmarks. Improve over existing methods by a small margin.
* Provide an extensive ablation study of the multi-level preference term.

Additionally, automating of the multi-level preference data generation could be a potential strength as well, but currently lacks evaluation to justify its quality (see weakness).

**Weaknesses:**

I would like to see authors address the following weaknesses:

* **Lack intrinsic evaluation of the automated multi-level preference dataset**. The quality is only implicitly justified by the improvement on the three final benchmarks (L258-L264), which makes it unclear what are the artifacts introduced in the automated data generation. Although human or GPT-4 annotation can be inconsistent sometimes, it is still good to collect some annotations to directly assess how the generated preferences align with the degree of hallucination. Similarly, the current auto-check mechanism is ad-hoc and introduces another component, i.e., CLIP, which could introduce additional errors into the system. It would be good to conduct some evaluation on the auto-check mechanism as well.
* **Missing comparison with rank-based preference alignment approaches**: Despite being a novel application, non-binary preference alignment has been studied both theoretically and empirically in context other than hallucination in MLLMs, for example Zhu et al. 2023 [1], Brown et al. [2], Myers et al. [3],  Song et al. [4]. It would be great if this work could engage with prior literature on non-binary preference alignment, for example, discussing how does the proposed objective compare with ranking-based approach in prior work?
* **Missing results of FGAIF on MRHal-Bench** In Table 1, FGAIF has a performance that is considerably close to the proposed methods (-0.14, +0.05) on MMHal-Bench and outperform the proposed method on LLaVA-Bench, yet it's missing results MRHal-Bench. These missing numbers could affect the comparison between the two methods.

References:
* [1] Zhu et al. Principled Reinforcement Learning with Human Feedback from Pairwise or K-wise Comparisons.
* [2] Brown et al. Safe imitation learning via fast bayesian reward inference from preferences.
* [3] Myers et al. Learning Multimodal Rewards from Rankings.
* [4] Song et al. Preference Ranking Optimization for Human Alignment.

**Questions:**

*  **Artifacts of the using responses from different model size**: Authors mentioned that inconsistent language styles can introduce biases, how does this concern justify the choice of using various responses from models of different sizes in the same model family? Responses from smaller models clearly don't just change the factual information, but also introduce more repetition and incoherence issues (for example, see Li et al. 2023 [1]). Would some simple perturbation-based methods control style and other factors better? The questions on artifacts apply to varying dataset size as well, it would be great if authors can discuss potential artifacts.
* **Why not use KL-Divergence for penalty** Author added a penalty term to avoid degrading the quality of the superior responses in formula (6), why use an entropy term instead of the standard KL-Divergence based shift penalty term in RLHF? Won't this penalty term allows reward hacking on the penalty?
* Minor: in formula (5), maybe the outer loop should be 0 to k-2?

[1] Li et al. Contrastive Decoding: Open-ended Text Generation as Optimization.

**Limitations:**

Yes.

---

> ### Author Rebuttal · Authors · 2024-08-07
>
> **Q1.** Lack intrinsic evaluation of the AMP dataset.
>
> **A1.** Thanks for your advice. We provide more evaluation for the AMP dataset and the auto-check mechanism.
>
> **Evaluation of the AMP dataset.** We estimate the inconsistency rate of our AMP dataset to be 2.25% (by manual evaluation on 2k random samples). The 2.25% inconsistency rate is significantly lower than the human/GPT-4V (14.40%/11.95% inconsistency) annotations. Below is an inconsistent case in our AMP dataset, where $N_i$ denotes the $i$-th noun chunks:
>
> >**Standard Response**: A little girl ($N_1$) with a purple jacket ($N_2$) is flying a kite ($N_3$).
> >
> >**Response A**: A little girl ($N_1$) dressed in purple jacket ($N_2$) is flying a kite ($N_3$) on the lawn, surrounded by *many people* (other information, including some *hallucinations* denoted by *italic*).
> >
> >**Response B**: A young girl ($N_1$) dressed in purple clothes ($N_4$, similar to $N_2$ but not exactly the same) is flying a kite ($N_3$).
>
> Response A accurately predicts these noun chunks and thus gets a higher score, leading to A>B. However, the actual ranking is B>A because A has some hallucinations.
>
> Moreover, we conduct another experiment to validate the superiority of our AMP dataset. We mix the AMP and human/GPT-4V data for training the model. Table 1 in the PDF document shows that as the proportion of human/GPT-4V annotated data increases, the hallucination rate rises accordingly.
>
> **Evaluation of the auto-check mechanism.** If we remove the auto-check mechanism of the sampled 2k examples, the inconsistency rate increases from 2.25% to 17.45%. It shows that the auto-check mechanism is critical.
>
> ---
>
> **Q2.** Missing comparison with rank-based preference alignment approaches.
>
> **A2.** We make both conceptual and empirical comparisons between our MDPO and prior rank-based preference alignment approaches, showing that our method has several advantages, $e.g.$, higher performance. The details are as below.
>
> **Conceptual comparison.** There are two main differences between our MDPO and other rank-based preference alignment approaches. First, our MDPO mitigates the challenge of distinguishing micro hallucinations in responses. Taking 3-level preference as an example, the comparisons of other methods are 'A>BC, B>C', while comparisons made by our AMP are 'A>B, A>C, B>C'. More specifically, our AMP splits 'A>BC' into 'A>B, A>C', which enables MLLMs to perceive the subtle differences between different responses. Second, our penalty term explicitly increases the probability of MLLMs generating good answers, ensuring the stability of the training process.
>
> **Empirical comparison.** As reported in Table 2, our MDPO surpasses these two learning objectives, $i.e.$, [1], [2] on all hallucination benchmarks, $e.g.$, 3.01 -> 3.17 on MMHal-Bench.
>
> ---
>
> **Q3.** Missing results of FGAIF on MRHal-Bench.
>
> **A3.** On MRHal-Bench, the mentioned FGAIF achieves 3.77/3.79 score and 0.30/0.31 hallucination rate. In comparison, our AMP is better, achieving 4.07/4.06 score (the higher the better) and 0.20/0.15 hallucination rate (the lower the better). Notably, the results of FGAIF are based on our re-implementation, because it is still close-sourced in terms of data and code. We are not confident that our implementation is completely correct. Once FGAIF is publicly available, we will update its results and add the comparison to our repo on Github.
>
> ---
>
> **Q4.** Artifacts of the using responses from different model size.
>
> **A4.** Using responses from different model sizes does have some artifacts, $i.e.$, some inconsistency. However, the inconsistency rate (2.25%) is significantly lower than the GPT-4V (11%) and human annotations (14%). Please kindly refer to the responses to Q1 for estimation details and examples.
>
> Moreover, we empirically validate that using same-family models is better than using different-family models. We build a dataset that mixes the responses from GPT-4O, Qwen-2 and LLaVA. Training model with this dataset (#3 in Table 2) achieves inferior results than our strategy, $e.g.$ , 2.89 (different family) versus 3.17 (ours) score on MMHal-bench.
>
> **Perturbation-based methods** also improve the baseline ($e.g.$, +0.14 on MMHal-Bench), but is still inferior to our MDPO ($e.g.$, -0.34 on MMHal-Bench), as shown in Table 2.
> Our implementation details are as follows. We randomly change the noun, adjective, preposition, and numeral. We obtain answers of varying quality by controlling the proportion of perturbations (10%, 30%, 50% based on 4-level preference setting). We infer the hallucination pattern generated by random perturbation is different from the real MLLM and is thus not informative enough for preference learning.
>
> ---
>
> **Q5.** Why not use KL-Divergence for penalty?
>
> **A5.** KL-Divergence is identical to our penalty term. The mathematical proof is as follows.
>
> Assume $\mathrm{P}(y_w)$ is the probability distribution of superior responses, $\pi_\theta(y_w|x)$ denotes the probability distribution of the model generating superior responses, KL-Divergence is:
> $$\mathbf{KL}[\mathrm{P}(y_w)||\pi_\theta(y_w|x)]=\sum_{y_w\in\mathcal{Y}} \left[\mathrm{P}(y_w) \log \frac{\mathrm{P}(y_w)}{\pi_\theta(y_w|x)} \right],\tag{1}$$
> where $\mathrm{P}(y_w)=[0,...,1,...,0]\in \mathbb{R}^{L}$ and $L$ is the length of vocabulary. Since $\mathrm{P}(y_w)$ is 0 at all positions except for the superior word, where Equ.1 is re-written:
> $$\mathbf{KL}[\mathrm{P}(y_w)||\pi_\theta(y_w|x)]=-\log\pi_\theta(y_w|x),$$
> $$\min\mathbf{KL}[\mathrm{P}(y_w)||\pi_\theta(y_w|x)]\Leftrightarrow\max\log\pi_{\mathrm{\theta}}\left(y_{w} \mid x\right).$$
> Therefore, our penalty term is actually KL-Divergence.
>
> ---
>
> **Q6.** Minor
>
> **A6.** The outer loop is 1 to K-2. We take the responses from the pre-trained MLLM as $R_0$.
>
> ---
>
> [1] Zhu et al. Principled Reinforcement Learning with Human Feedback from Pairwise or K-wise Comparisons.
>
> [2] Song et al. Preference Ranking Optimization for Human Alignment.

---

> > ### Comment · Reviewer_2sCy · 2024-08-12
> >
> > Thanks for the detailed responses and the additional experiment, which addressed my concerns. I have raised the score accordingly.

---

### Author Rebuttal · Authors · 2024-08-07

We would like to express our sincere gratitude to the reviewers for their valuable comments. We are encouraged they found our method is "**novel**" (Reviewer 2sCy), our method "**provides a broader range of comparisons**" (Reviewer 2sCy, Reviewer iDRq), our automated pipeline is a "**significant contribution, potentially reducing bias and noise while saving time and resources**" (Reviewer xo7T). To address all the reviewers’ concerns, we provide experiments, discussions, and point-to-point responses. We will add these experiments and discussions in the final version.

Tables mentioned in the rebuttal answers are all in the PDF document. We provide Table 1 for **Reviewer 2sCy, Reviewer iDRq**, Table 2 for **Reviewer 2sCy**, Table 3 for **Reviewer iDRq**, Table 4 for **Reviewer iDRq**. The details of tables are listed as follows:

---

**Table 1.** Performance on three hallucination benchmarks across different proportions of GPT-4V/human annotations.

To assess how the inconsistency aligns with the degree of hallucination, we incorporate human/GPT-4V annotations into our automated dataset based on the 4-level setting (the optimal setting).

**Implementation Details.** On our training dataset, the ratio of contradictory patterns in human/GPT-4V annotations is 16.9%/14.4%. To get preferences from contradictory patterns, we ask human/GPT-4V to annotate all levels directly instead of annotating them pairwise. Then, we integrate the human/GPT-4V annotated dataset into our automated dataset, with proportions of 70%, 50%, 30%, and 10%, respectively.

**Results.** As the proportion of human/GPT-4V annotated data increases, the hallucination rate rises accordingly, which proves the advantage of our automated preference dataset.

---

**Table 2.**
Performance on three hallucination benchmarks across other perturbation-based methods (\#1, \#2), MLLMs from different families (\#3, DF), perturbation-based methods (\#4, PB), our baseline (\#5), and our MDPO.

---

**Table 3.**
Performance of LLaVA-V1.6, DeepSeek-VL, and MiniCPM-V on three hallucination benchmarks.

---

**Table 4.**
Effectiveness of loss for BC.

---

### Decision · Program_Chairs · 2024-09-25

**Decision:**

Accept (poster)

**Comment:**

This submission seeks to mitigate hallucinations in Multimodal Large Language Models by optimizing preferences. Motivated by the shortcomings of binary preferences, the authors introduce a multi-level preference framework.

We appreciate the responses from authors and the discussions with the reviewers. Please add those experiments and discussions in the final version.